# Optical Biopsy of Dysplasia in Barrett’s Oesophagus Assisted by Artificial Intelligence

**DOI:** 10.3390/cancers15071950

**Published:** 2023-03-24

**Authors:** Jouke J. H. van der Laan, Joost A. van der Putten, Xiaojuan Zhao, Arend Karrenbeld, Frans T. M. Peters, Jessie Westerhof, Peter H. N. de With, Fons van der Sommen, Wouter B. Nagengast

**Affiliations:** 1Department of Gastroenterology and Hepatology, University Medical Center Groningen, University of Groningen, 9713 GZ Groningen, The Netherlands; 2Department of Electrical Engineering, Video Coding and Architectures, Eindhoven University of Technology, 5612 AZ Eindhoven, The Netherlands; 3Department of Pathology, University Medical Center Groningen, University of Groningen, 9713 GZ Groningen, The Netherlands

**Keywords:** Barrett’s dysplasia, surveillance, computer-aided diagnosis, machine learning, medical training, endocytoscopy

## Abstract

**Simple Summary:**

Advanced endoscopy techniques that generate microscopic images can be used to optimize cancer screening in patients with an increased risk of oesophageal cancer. However, these microscopic endoscopy images are highly detailed and difficult for doctors to interpret. Support by artificial intelligence (AI) could be useful when the image is too complex for human interpretation. Therefore, this study investigated whether AI as second assessor can assist doctors in assessing complex, microscopic endoscopy images for the presence of oesophageal cancer. To investigate this, we developed online training and testing modules for doctors to learn to classify these novel images, and to assess the potential of AI assistance in analysing the oesophageal microscopy images. Our data showed that the best diagnostic scores for cancer recognition emerged through the collaboration between doctors and AI as the second assessor. Therefore, AI could be used to support the clinical implementation of endoscopy techniques that generate microscopic images.

**Abstract:**

Optical biopsy in Barrett’s oesophagus (BE) using endocytoscopy (EC) could optimize endoscopic screening. However, the identification of dysplasia is challenging due to the complex interpretation of the highly detailed images. Therefore, we assessed whether using artificial intelligence (AI) as second assessor could help gastroenterologists in interpreting endocytoscopic BE images. First, we prospectively videotaped 52 BE patients with EC. Then we trained and tested the AI pm distinct datasets drawn from 83,277 frames, developed an endocytoscopic BE classification system, and designed online training and testing modules. We invited two successive cohorts for these online modules: 10 endoscopists to validate the classification system and 12 gastroenterologists to evaluate AI as second assessor by providing six of them with the option to request AI assistance. Training the endoscopists in the classification system established an improved sensitivity of 90.0% (+32.67%, *p* < 0.001) and an accuracy of 77.67% (+13.0%, *p* = 0.020) compared with the baseline. However, these values deteriorated at follow-up (−16.67%, *p* < 0.001 and -8.0%, *p* = 0.009). Contrastingly, AI-assisted gastroenterologists maintained high sensitivity and accuracy at follow-up, subsequently outperforming the unassisted gastroenterologists (+20.0%, *p* = 0.025 and +12.22%, *p* = 0.05). Thus, best diagnostic scores for the identification of dysplasia emerged through human–machine collaboration between trained gastroenterologists with AI as the second assessor. Therefore, AI could support clinical implementation of optical biopsies through EC.

## 1. Introduction

Patients with Barrett’s oesophagus (BE) are at an increased risk of developing oesophageal adenocarcinoma (EAC), which can evolve from low-grade and high-grade dysplasia (LGD/HGD) [1]. Detection of early-stage EAC and its dysplastic stages is key to providing the patient with endoscopic treatment with excellent prognosis [2,3]. Therefore, BE patients are subject to endoscopic surveillance examinations at stratified intervals with four-quadrant random biopsies every 1–2 cm along the Barrett segment (the Seattle protocol) [4,5]. Despite high-definition white light endoscopy (HD WLE) and narrow-band imaging (NBI) highlighting the vascular patterns, dysplasia in BE can be hard to discriminate endoscopically from non-dysplastic metaplastic tissue, as they present solely as subtle morphological changes [6]. Furthermore, the sampling protocol is associated with sampling error, with reported missed dysplasia rates in a quarter to half of the patients with inconspicuous neoplasia [7,8], which especially affects patients with a long BE segment (>10 cm) [9]. Overall, a 3.1% dysplasia detection rate is achieved during surveillance examinations [10]. Hence, despite the labour-intensive and time-consuming characteristics, many samples can be obtained with a relatively low yield for dysplasia or cancer during surveillance endoscopy of BE.

Endocytoscopy (EC) is an advanced endoscopy technique that combines HD WLE and NBI with ultra-magnification technology. The fourth-generation endocytoscope (GIF-H290EC, Olympus, Tokyo, Japan) enables sequential evaluation of tissue ranging from an HD WLE overview to microscopic inspection with a magnification factor of 520× for real-time visualization of the histology and cytology of the superficial mucosa [11]. Consequently, EC can potentially generate an “optical biopsy” during endoscopy. However, the current experience with EC in BE for this purpose is limited due to the technical insufficiencies of the previous EC models to generate adequate in vivo imaging [12,13]. As previously observed with another ultra-magnification endoscopy technique, confocal laser endomicroscopy (CLE), this approach could reduce the number of tissue samples and save a considerable amount of time compared with the current BE sampling approach [14,15]. Consequently, the associated costs could be reduced as well [16]. However, to optimally exploit the functionality of optical biopsies, substantial training and experience are required for the operator, who must pass the Preservation and Incorporation of Valuable Endoscopic Innovations (PIVI) threshold values for potential clinical implementation [17,18]. As a result, practicing gastroenterologists are unlikely to adopt these techniques [19,20].

To improve the assessment and processing of the information that is provided by endoscopy, artificial intelligence (AI) algorithms are increasingly being developed. In recent years, a form of AI called deep learning—more specifically, convolutional neural networks (CNN)—has been extensively studied for analysing medical imaging data [21]. AI has the potential to help endoscopists with real-time interpretation of the increasingly complex information from advanced endoscopies and with the associated clinical decisions [22]. However, previous research has focused mainly on the role of AI as an independent primary assessor of endoscopic imaging, most commonly in a human-vs.-machine setup. Nevertheless, in future clinical practice, AI probably has much more potential in a complementary role to the endoscopist as a second assessor, in which it could aid in the interpretation of these complex images [23]. Subsequently, AI could be used to assist gastroenterologists with the optical biopsy of dysplasia in complex images via advanced endoscopy techniques to alleviate the burden of training and experience, and to ensure efficient implementation of optical biopsies by novel imaging modes in clinical practice. Therefore, data regarding human–machine interactions are required, as the confidence of gastroenterologists with AI will affect the impact of AI on the introduction of optical biopsies in endoscopy.

In this study, we aimed to establish the potential of AI as a second assessor to help gastroenterologists in identifying dysplasia through an optical biopsy in BE generated by EC. Therefore, we first prospectively acquired EC frames of BE. We used these frames to develop and validate an endocytoscopic BE classification system in a cohort of 10 endoscopists through an online platform, and to train and test a CNN architecture. The platform consisted of online modules that included a training program and three test sets: a pretraining test (Image Set 1A), a post-training test (Image Set 1B), and a follow-up test (Image Set 2). After the pilot phase of the study, a second cohort of 12 gastroenterologists completed the same online modules: one group of six gastroenterologists without AI support and one group of six gastroenterologists who had access to feedback from the CNN if they requested it. This approach enabled us to compare the performances of unassisted and AI-assisted gastroenterologists in interpretating endocytoscopic BE images and to evaluate the human–machine interactions.

## 2. Materials and Methods

### 2.1. Study Setting and Population

BE patients referred to the gastroenterology and hepatology department of the University Medical Centre Groningen (UMCG) were prospectively imaged with EC (model GIF H290-EC) to acquire endocytoscopic BE frames for the purpose of the study from May 2019 to September 2020. The procedure that we used to perform in vivo EC videotaping and to obtain targeted biopsies is described in Appendix A.

All participants received oral and written information before informed consent was obtained. Patients were eligible when they were 18 years or older and scheduled for surveillance endoscopy of their BE or endoscopic removal of their Barrett’s neoplasia. The trial was approved by the UMCG’s institutional review board and was registered in the Dutch Trial Register (NL8573).

### 2.2. Online Platform with Training and Testing Modules

We designed online modules in REDCap (Research Electronic Data Capture) to train and test gastroenterologists in interpretating our proposed endocytoscopic BE classification system. The online training and testing with AI could be accessed via https://redcap.link/Endocyto_AI_test-training. The four-step process of developing our endocytoscopic BE classification system is reported in Appendix A, which also includes a detailed explanation of the online training and testing modules.

#### 2.2.1. Participants in the Online Modules

Participants in the online modules were recruited on the basis of their time to respond to the invitation: the ones that responded first to the invitation were included as participants. For the pilot phase, the first cohort of 10 endoscopists (five gastroenterology residents and five gastroenterologists with expertise in BE) was recruited from 15 invitees to participate in the online training and testing modules (Figure 1A). Afterwards, we randomly recruited 12 additional gastroenterologists from our centre and from general hospitals for online participation from a group of 18 invitees: one group of six gastroenterologists participated in the tests without AI support, the other group had the versions of the test sets with AI support (Figure 1B). Lastly, a cross-over was performed: the six gastroenterologists that initially completed the version with Test Set 2 without AI support were requested to take the test with Set 2 again six weeks later, but this time, they used the version of Test Set 2 that had the option of calling for AI support. All participants did not have any experience in the endoscopic application of AI or in EC. The baseline characteristics of the online participants can be found in Appendix A.

#### 2.2.2. Training and Testing Modules

In the online training program for the gastroenterologists, we explained the EC procedure (Figure 2A) and the features of our endocytoscopic BE classification system (Figure 2B) through multiple images and (narrated) videos. We developed a binary system that differentiated EC metaplasia, including histologically verified metaplastic columnar tissue of the fundic type, cardia type, or intestinal type, from EC neoplasia, including histologically verified LGD, HGD, and carcinoma (for the features of the proposed endocytoscopic BE classification system, see Appendix A). Test Set 1 was completed by the gastroenterologist before (Test Set 1A) and after training (Test Set 1B), with the same images in different sequences. An invitation to Test Set 2 was sent after a two-week interval (Figure 1). The gastroenterologist diagnosed the image as either EC metaplasia or EC neoplasia and indicated the confidence level (high or low) of their decision. The test sets were all designed with two versions: one version without the implementation of AI support, and one version in which the gastroenterologists could request a prediction from the AI model. In the latter version, the participant was first presented with an EC image and asked whether he/she wanted AI support (Figure 3A). If the answer was “yes”, then the image was shown with a prediction of the diagnosis by the AI algorithm. The predicted diagnosis was displayed as a median value with IQRs on a continuous scale (from 0 at the green end (EC metaplasia) to 100 at the red end (EC neoplasia)). The median value was also displayed as a score in the bottom left corner. The participants with the version of the test sets with AI support were aware of the sensitivity, specificity, and accuracy of the AI algorithm for the separate test sets.

### 2.3. Training, Validating, and Testing of the CNN Architecture

For the AI model, we used the decoder of a CNN architecture based on ResNet, which has been used in previous oesophageal CAD work [24]. The model was pretrained on the ImageNet dataset [25], and then trained and tested using data that were extracted from endocytoscopic BE videos (Figure 3B). The data were preprocessed by resizing to a resolution of 512 by 512 pixels and by converting the image to grayscale. This step was performed to avoid bias in a particular colour staining, since the number of available patients was relatively small. Fivefold patient-based cross-validation was used to optimize the algorithm, thereby training a separate model for each fold, which was subsequently used as an ensemble on the test set. It should be emphasized that the training and validation data were strictly separated on the patient level from the testing data to avoid overfitting. To further improve generalization of the models, data augmentation was used. Images were randomly flipped over the x- and y-axes with *p* = 0.5, and randomly rotated by a multiple of 90°. Additionally, the images were translated by up to 10% of the image’s width and sheered by a maximum of 8°. Finally, a random crop of 256 by 256 pixels was extracted during the training phase. In the testing phase, no augmentations were performed apart from resizing and converting the images to grayscale. A cyclic cosine annealing learning rate scheduler [26] was used in combination with Adam and AMS-grad [27] to control the learning rate. Binary cross-entropy was used as the loss function. The five models trained using fivefold cross-validation were then ensembled to obtain a final classification score per image.

For the construction of the dedicated datasets for training and testing the AI model, we first reviewed endocytoscopic BE videos (see Appendix A). We excluded videos that were of a low quality due to the lack of image resolution and/or insufficient staining of the mucosa (N = 17) (Appendix A). High-quality videos in which the corresponding targeted biopsies were indefinite for dysplasia (N = 2) were also excluded. The remaining 59 endocytoscopic videos were used to extract frames, resulting in a total of 83,277 frames. Frames that were unrecognizable due to the scattering and reflection of light, a lack of camera focus, or peristaltic and respiratory movements were excluded from the selection. To achieve heterogeneity in the morphology of the frames, one or two frames were manually selected per 25 frames from each video sequence. To create distinct datasets, the selected frames of patients were split on a per-patient basis into either the training set or the test sets. Some patients contributed EC metaplasia frames as well as EC neoplasia frames to a dataset. Subsequently, the AI algorithm was trained using a dataset of 1552 frames that was selected from 35 videos of 20 patients. The two test sets were composed of an independent set of 710 frames that was selected from 24 videos in another 17 patients and were the same test sets that were completed by the gastroenterologists participating online.

### 2.4. Statistical Analysis

To assess the potential of AI as a second assessor to help gastroenterologists in analysing endocytoscopic BE images, we used an accuracy of 85%, which was drawn from the sensitivity (90%) and specificity (80%) thresholds of the PIVI guidelines for potential clinical implementation of new endoscopy techniques, and the follow-up accuracy (70%) of the endoscopists during the pilot phase, which was significantly lower than their post-training accuracy. A power of 80% and a two-sided test level at α = 0.05 were assumed to demonstrate a difference of 15% between the accuracy of the gastroenterologists with AI and the accuracy of gastroenterologists without AI after a two-week post-training interval. Thus, we included two groups of six gastroenterologists to establish 360 observations per testing moment.

Diagnostic performance was calculated on a per image basis using the values of sensitivity, specificity, and accuracy (with 95% CIs) and with the histology as a reference standard. Categorical variables were presented as numbers and proportions, and continuous variables as medians (IQR) or means (SD), depending on the distribution. We compared the proportions through the *χ*^2^-test or Fisher’s exact test, means through the appropriate T-test or one-way ANOVA, and medians through the Mann–Whitney U-test. Here, *p* < 0.05 (two-sided) was considered statistically significant for all analyses. To assess the level of interobserver agreement, kappa values were calculated and interpreted according to the arbitrary thresholds of Landis and Koch [28]. Statistical analysis was performed using SPSS Statistics 27 (IBM, Armonk, New York, NY, USA).

## 3. Results

### 3.1. Validation of the Endocytoscopic BE Classification System by Endoscopists

Compared with their baseline performance, the post-training performance of the endoscopists improved regarding their sensitivity for recognizing dysplasia by 32.67% (95% CI 23.28–42.06, *p* < 0.001) and accuracy with 13.0% (2.55–23.45, *p* = 0.020) (Table 1). After an interval of 30.4 (±16.0) days, their sensitivity and accuracy dropped by 16.67% (9.12–24.21, *p* < 0.001) and 8.0% (2.59–13.41, *p* = 0.009) respectively. Nevertheless, the sensitivity at follow-up remained higher than at baseline (+16.0% (5.65–26.35), *p* = 0.007). Comparably, the interobserver agreement scores for the endoscopists were moderate when tested before training (0.482, 95% CI: 0.450–0.514) and at follow-up (0.492, 0.459–0.525), but substantial (0.646, 0.610–0.681) when tested directly after training on the classification system. An overview of the experiences of the endoscopists regarding the online modules are included in Appendix A.

### 3.2. Results of Testing the AI Algorithm

The ROC curves that were obtained as an average from the five CNN models resulted in AUCs of 0.894 for Test Set 1 and 0.923 for Test Set 2.

### 3.3. AI-Assisted Gastroenterologists versus Unassisted Gastroenterologists

For the sensitivities, specificities, and accuracies of the gastroenterologists with and without AI support, see Table 2. When tested at 23.3 (±9.1) days after training, AI-assisted gastroenterologists maintained higher sensitivity (+20.0% (3.14–36.86), *p* = 0.025) and accuracy (+12.22% (0.03–24.48), *p* = 0.050) for recognizing dysplasia than the unassisted gastroenterologists. Furthermore, all the kappa values of the unassisted gastroenterologists were equivalent to moderate, whereas both post-training scores (Test Sets 1B and 2) of the AI-assisted gastroenterologists were substantial (Table 3). In Figure 4A–C, an overview of the ROC curves of the CNN and the scores of the AI-assisted and unassisted gastroenterologists are plotted per testing moment.

The proportions of true diagnoses that were established with a high confidence level were higher among gastroenterologists with AI support than among gastroenterologists without AI support before training (32.8% vs. 11.7%, *p* < 0.001), after training (57.2% vs. 43.9%, *p* = 0.011), and at follow-up (53.9% vs. 35%, *p* < 0.001). An overview of the experiences of the gastroenterologists regarding the online modules are included in Appendix A.

### 3.4. Cross-Over of Unassisted Gastroenterologists to AI Assistance

After an interval of 54.5 (±10.4) days, the gastroenterologists without AI support attempted Test Set 2 again, this time with AI support, exceeding their previous sensitivity (+24.45% (95% CI 5.85–43.04), *p* = 0.020) and accuracy (+13.33% (1.02–25.65), *p* = 0.039) without AI support. Additionally, they outperformed the sensitivity of the AI model alone (+12.89% (2.11–23.67), *p* = 0.024) (Figure 4D). Overall, gastroenterologists opted for AI support in 64.4% (116/180) of the cases during this test. Interestingly, all six gastroenterologists were able to identify metaplastic tissue in one specific EC image after calling for AI support, despite an indecisive AI prediction.

### 3.5. Human–Machine Interactions

Gastroenterologists who had access to AI support called for AI support in 90% (162/180) of the cases before training, which dropped to 57.8% (104/180) after training (*p* < 0.001). At follow-up, 75.6% (136/180) called for AI support, which was still less than before training (*p* < 0.001). Notably, the trained gastroenterologists made fewer false negative diagnoses in images that otherwise would have been correctly predicted by AI (6.7% (3/45) vs. 37.5% (3/8), *p* = 0.038). Furthermore, we observed the following two tendencies. First, true positive diagnoses predicted by AI were erroneously rejected by trained gastroenterologists in 3.8% (4/105) of the cases and in 7.9% of cases (6/76) by the untrained gastroenterologists (*p* = 0.325). Second, correct diagnoses by the gastroenterologists for images with an indecisive AI prediction were 50% (6/12) before training and 77.8% (14/18) after training (*p* = 0.118). In Appendix A, data regarding the human–machine interactions of the gastroenterologists that had access to AI support are provided for all test sets.

## 4. Discussion

In this study, we demonstrated that AI as a second assessor enhances the performance of gastroenterologists who have been trained to identify dysplasia in optical biopsies of BE generated by EC. Notably, trained gastroenterologists with AI support achieved sensitivities of approximately 95%, although their specificities remained suboptimal. Through our online platform, we saw that AI-assisted gastroenterologists scored substantially better than unassisted gastroenterologists in establishing an optical biopsy of Barrett’s-associated dysplasia through EC. Additionally, gastroenterologists appeared to be more confident in interpreting this novel imaging when they had access to AI support, as shown by the larger proportion of high-confidence diagnoses in this group compared with gastroenterologists without AI support. Lastly, the AI-assisted gastroenterologists achieved generally higher kappa values for interobserver agreement than the gastroenterologists without AI support, suggesting that access to AI facilitated a more objective interpretation of the EC features of BE.

Because the level of visual detail in EC is increased compared with conventional HD WLE, its information is considered complex and results in a more challenging real-time interpretation. As no in vivo endocytoscopic BE classification system was available, we developed a classification system that enabled the discrimination of dysplastic from non-dysplastic BE during a pilot study. After 10 endoscopists were trained online to use this system, their diagnostic scores increased compared with the pretraining scores. However, the diagnostic performances deteriorated at follow-up, indicating that the interpretation of this novel imaging was not straightforward and AI might be a necessary supplement. Previous studies using AI in EC applications have not been conducted for BE but have predominantly focused on colorectal polyps [29,30,31,32]. In one study, a CNN outperformed 30 endoscopists by distinguishing neoplastic from benign polyps from the EC imagery with sensitivities, specificities, and accuracies of up to 96.9%, 100%, and 98%, respectively [33]. These results suggested that optical biopsy of colorectal lesions is possible; based on our results, we believe that this also applies to BE.

In the past, experience with endoscopic microscopy imaging for the purpose of optical biopsy in BE has been gained through CLE. Although CLE generally achieved promising diagnostic test characteristics in BE [34,35,36,37], it has not been implemented into routine clinical practice in most centres. As CLE is a probe-based technique, it needs to be inserted in the working channel of the endoscope, which hampers the operator in performing a targeted biopsy after imaging has been performed, potentially inducing sampling errors. The current endoscope-based EC model allowed the screening of BE in a continuous manner after competence had been developed, as well as direct sampling of the aberrant tissue, as a biopsy forceps can be inserted through its working channel during EC imaging. Nevertheless, EC remains a technically challenging procedure, especially in BE, but could be promising in surveillance of Barrett’s oesophagus, especially with AI assistance, and future real-time studies may show its benefit over the standard Seattle biopsy protocol.

As we introduced EC for in vivo use in BE, a novel type of imaging was generated that consequently lacked clinically standardized interpretations. Therefore, the development of our classification system was crucial. Without this reference standard, the gastroenterologists would not have any tools for interpreting these novel images, thus risking excessive reliance on AI [38]. Indeed, we observed this behaviour among the untrained gastroenterologists who had access to AI: the rates of calling for AI support after training and at follow-up were both significantly lower than before training. After they were able to independently interpret the imagery, the gastroenterologists were more likely to acknowledge a correct diagnosis made by AI and compensate for the outliers that were not recognized by AI. Therefore, the best diagnostic performances were achieved by combining the abilities of trained gastroenterologists with the AI model. Comparable observations for human–machine collaboration have also been made in the fields of dermatology and radiology [39,40].

Various AI algorithms have been developed for interpreting endoscopic images in BE, but none have yet been adopted in clinical practice. The implementation of AI in endoscopic surveillance for lesions in BE is thought to be beneficial, as it has shown its superiority in detecting and demarcating neoplastic lesions in conventional endoscopic imaging compared with non-expert endoscopists [41]. However, the human factor should be taken into account for the exploitation of AI-driven endoscopic applications. We therefore suggest that studies investigating AI in GI endoscopy should change their experimental designs from human vs. machine to human-with-machine vs. human-without-machine.

This study had several limitations. As we generated a unique in vivo endocytoscopic high-quality BE dataset of 520-fold magnified images, we were unable to test the performance of the CNN with external datasets. Although we strictly split our cohort on a per-patient basis into training and testing datasets, it should be acknowledged that the algorithm worked well in this particular cohort but that its performance remains unknown for the out-of-distribution data. Due to the relatively small dataset, little variation was present in the imaged tissues, which, in turn, limited the diversity of EC imagery that could be incorporated in the dedicated CNN training set and shown to the gastroenterologists in their online training program. The specificity scores could therefore have been affected. However, we expect that the promising performance of the AI model in the present study will improve when the training set is enlarged. Furthermore, the online environment did not fully reflect clinical practice. We chose this deliberately, as we wanted the gastroenterologists to remain unbiased when assessing the EC imagery. In the online modules, they assessed only a single static EC image of the tissue and did not have information from the non-magnified images of the area. In clinical practice, however, endoscopists gather information about a particular area through the observations by HD WLE prior to EC assessments. If the approach used in the present study can be transferred to the endoscopy suite and the AI training set could be enlarged, we would expect increased specificity and a reduction in unnecessary biopsies, as a better selection of tissue would be offered to the CNN. Although the currently used AI algorithm is an image-based application, we expect that this approach could be conveniently used in real-time assessments of videos. As EC visualizes tissue at a 520-fold magnification during lens–mucosa contact, the frames from a video sequence are relatively similar, especially in comparison with HD WLE overviews, where various angles and distances between the lens and mucosa intervene in analyses of the images. EC frames with insufficient image quality due to artefacts could be excluded from the analysis by a supportive model [42,43]. Thus, the translation to real-time video assessments would be feasible and would potentially enhance the AI’s performance, as the outliers could be filtered out by averaging over multiple frames. Subsequently, the AI algorithm could generate more stable and robust predictions. Since the output of our algorithm was on a continuous scale, eventual variation in the probability of the predicted diagnosis will remain comprehensible during a dynamic assessment and without frequent switching of the prediction when using a categorical output. Moreover, in our approach, AI served only as a second assessor, and the human operator could decide whether to consult it or not as an artificial assistant for image interpretation.

A strength of this study is that we took an important concept into consideration in the design of our online examination modules for clinical decision-making with AI assistance: the human in the loop approach [44]. This approach is a branch of AI that leverages both human and machine intelligence to create machine learning models. In AI for medical imaging, such as the detection and characterization of lesions in GI endoscopy, an expert physician should still make the final decisions and assume responsibility, regardless of whether help from a computer-aided system is available or not. Therefore, clinicians should be educated about the limitations and strengths of the output of deep learning networks [45]. We believe that creating online resources for training and testing clinicians in using AI are indispensable for this education, which will enable the smooth integration of deep learning techniques in future clinical practice.

## 5. Conclusions

In conclusion, the collaboration between trained gastroenterologists and AI as a second assessor enhanced the recognition of dysplasia in endocytoscopic BE images, resulting in better scores than those of AI or human assessors alone. Additionally, AI could reduce the amount of training and clinical experience that is needed to interpret complex images, such as EC images of BE. As the imaging quality of endoscopy improves, the level of visual detail is enhanced, making the interpretation of advanced endoscopy techniques less straightforward for gastroenterologists. As well as reducing the time and effort that is required by gastroenterologists to master the assessment of imagery from advanced endoscopy, AI will also help to realize its potential benefits, including in optical biopsies. These factors would make this approach interesting for optimizing the diagnostic yield of surveillance examinations of BE. Therefore, future clinical implementation of optical biopsies by advanced endoscopy techniques could be facilitated when these systems are released with a corresponding AI algorithm that serves as a second assessor to help the gastroenterologist.

## Figures and Tables

**Figure 1 cancers-15-01950-f001:**
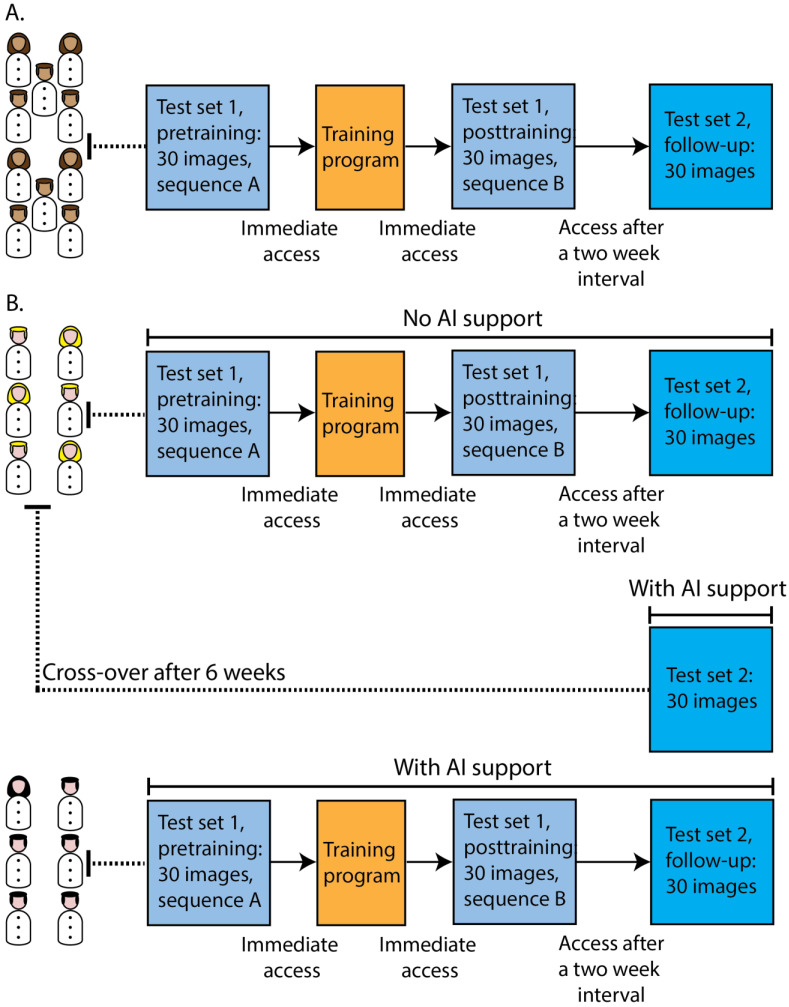
Schematic overview of the two cohorts participating in the online training and testing modules. (**A**) The first cohort of 10 endoscopists participated in the online training and testing modules during the pilot phase to validate the classification system. (**B**) A second cohort of 12 gastroenterologists participated in the same online training and testing modules: six unassisted gastroenterologists and six AI-assisted gastroenterologists. Lastly, a cross-over was performed in which the unassisted gastroenterologists were presented with Test Set 2 again, this time with AI assistance.

**Figure 2 cancers-15-01950-f002:**
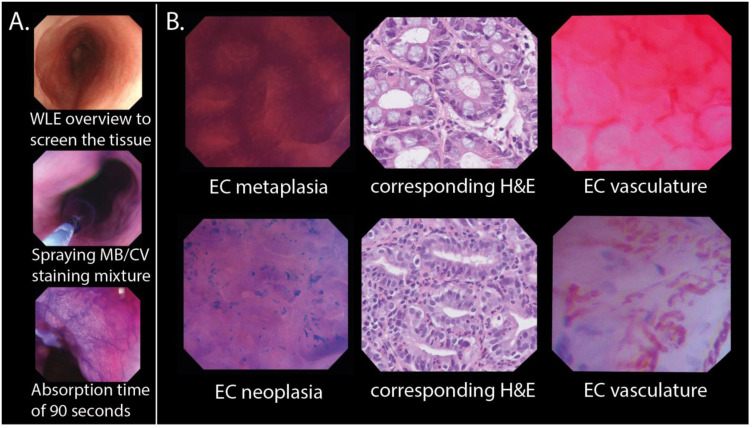
In vivo EC procedure and classification. (**A**) Sequential assessment of the BE performed during endoscopic examination using EC. (**B**) Examples of EC metaplasia and EC neoplasia with their corresponding histopathology. Examples of EC images showing microvascular features in metaplastic and neoplastic tissue are shown to the right. The EC metaplasia and EC neoplasia images were used to create endocytoscopic BE datasets. BE: Barrett’s oesophagus; CV: crystal violet; EC: endocytoscopy; H&E: haematoxylin and eosin staining; MB: methylene blue; WLE: white light endoscopy.

**Figure 3 cancers-15-01950-f003:**
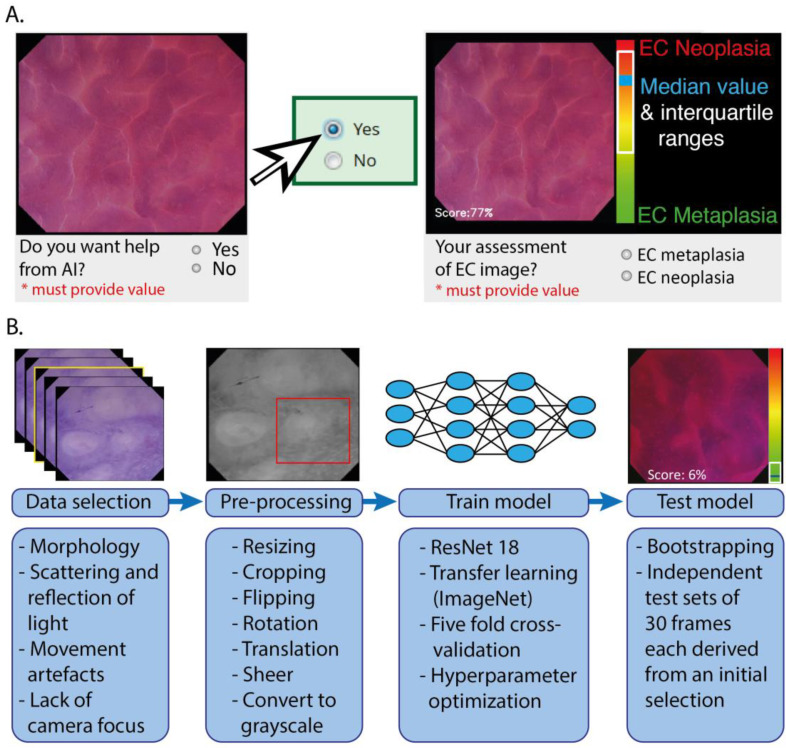
Overview of the display and the formation of the output of the CNN architecture. (**A**) After a request, AI assistance was presented as a score on a continuous scale (0–100), which was the average score of all five models in the ensemble. (**B**) To develop a CNN architecture for the interpretation of endocytoscopic BE images, we trained and tested in distinct image sets of preprocessed images that were selected from the prospectively acquired dataset.

**Figure 4 cancers-15-01950-f004:**
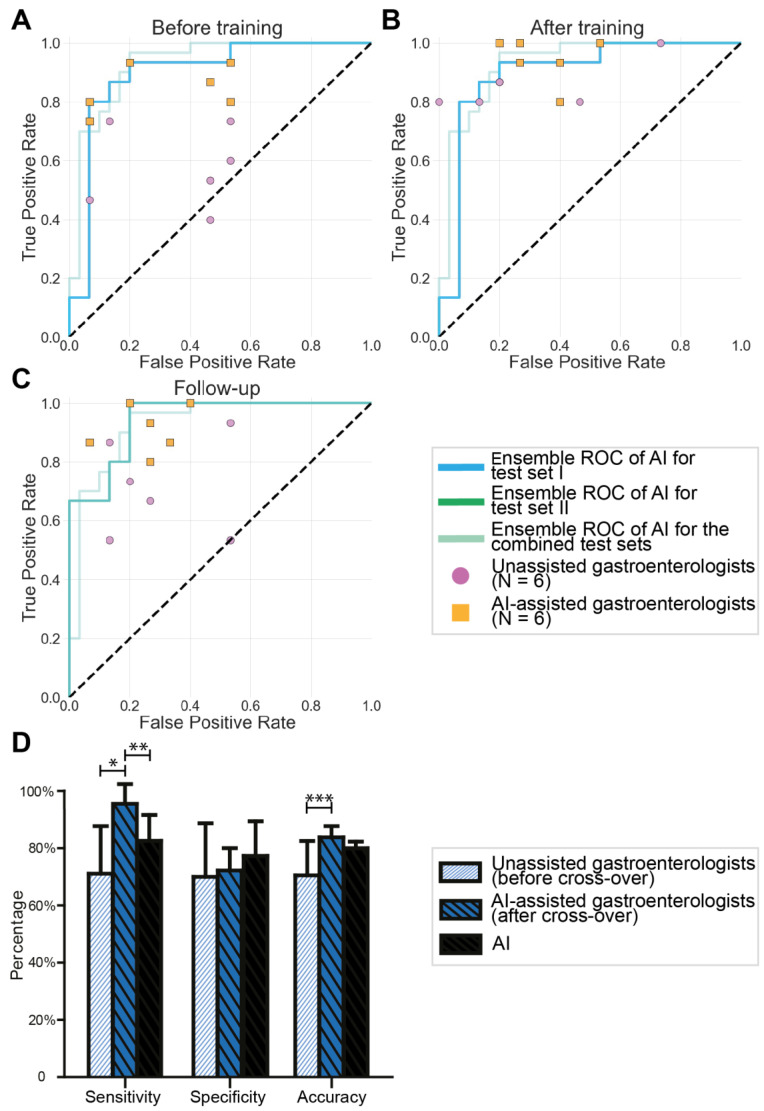
Diagnostic performance of unassisted gastroenterologists and AI-assisted gastroenterologists during the online modules. (**A**–**C**) Overview of the gastroenterologists with AI support (N = 6) and without AI support (N = 6) before training (**A**), after training (**B**), and at follow-up (**C**) relative to the overall AI ROC curves for Test Set 1 and Test Set 2, and both test sets combined. (**D**) Sensitivity, specificity, and accuracy scores of the unassisted gastroenterologists for Test Set 2 before the cross-over and their scores for Test Set 2 with AI support after the cross-over, and the scores of the AI model alone for Test Set 2. * *p* = 0.020, ** *p* = 0.024, *** *p* = 0.039.

**Table 1 cancers-15-01950-t001:** Diagnostic test scores of the endoscopists at various testing moments.

	Test Set 1		Test Set 2
	Before Training (Test Set 1A)	After Training (Test Set 1B)	Follow-Up
Sensitivity (95% CI)	57.33% (48.86–65.80)	90.0% (85.37–94.63)	73.33% (63.03–83.84)
Specificity (95% CI)	72.0% (57.47–86.53)	65.33% (56.40–74.27)	66.0% (56.09–75.92)
Accuracy (95% CI)	64.67% (55.12–74.22)	77.67% (73.03–82.31)	69.67% (64.71–74.62)

**Table 2 cancers-15-01950-t002:** Diagnostic test scores of unassisted and AI-assisted gastroenterologists at various testing moments.

		Test Set 1				Test Set 2	
		Before Training (Test Set 1A)	*p*-Value	After Training (Test Set 1B)	*p*-Value	Follow-Up	*p*-Value
Sensitivity (95% CI)	Unassisted gastroenterologists	57.78% (43.33–72.23)	0.002	85.56% (77.38–93.73)	0.076	71.11% (53.60–88.63)	0.025
	AI-assisted gastroenterologists	84.44% (76.97–92.91)		94.44% (86.26–100)		91.11% (82.64–99.58)	
Specificity (95% CI)	Unassisted gastroenterologists	63.33% (41.32–85.35)	0.668	71.11% (43.24–98.97)	0.652	70.0% (50.34–89.66)	0.631
	AI-assisted gastroenterologists	68.89% (45.20–92.58)		65.56% (52.72–78.39)		74.44% (62.93–86.50)	
Accuracy (95% CI)	Unassisted gastroenterologists	60.56% (47.58–73.54)	0.033	78.33% (67.11–89.56)	0.765	70.56% (57.96–83.15)	0.050
	AI-assisted gastroenterologists	76.67% (66.05–87.28)		80.0% (71.72–88.28)		82.78% (76.36–89.20)	

The *p*-values were calculated for separate comparisons regarding the sensitivity, specificity, and accuracy between unassisted gastroenterologists (N = 6) and AI-assisted gastroenterologists (N = 6) at every moment of testing (before training, directly after training, and at follow-up).

**Table 3 cancers-15-01950-t003:** Interobserver agreement for interpretation of endocytoscopic BE images for unassisted and AI-assisted gastroenterologists at various testing moments.

		Test Set 1		Test Set 2
		Before Training (Test Set 1A)	After Training (Test Set 1B)	Follow-Up
Kappa values (95% CI)	Unassisted gastroenterologists	0.491 (0.437–0.545)	0.562 (0.501–0.623)	0.526 (0.469–0.583)
	AI-assisted gastroenterologists	0.597 (0.537–0.657)	0.687 (0.622–0.752)	0.696 (0.631–0.761)

Interpretation of the interobserver kappa values according to the thresholds of Landis and Koch: scores less than 0.00: poor; 0.00 to 0.20: slight; 0.21 to 0.40: fair; 0.41 to 0.60: moderate; 0.61 to 0.80: substantial; 0.81 to 1.00: almost perfect [28].

## Data Availability

The data that support the findings of this study are available from the corresponding author upon reasonable request.

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
