# Peer review of "Optical Biopsy of Dysplasia in Barrett’s Oesophagus Assisted by Artificial Intelligence"

_cancers, 2023, doi:10.3390/cancers15071950_

Round 1

Reviewer 1 Report

This is an interesting study exploring the possibility of applying intelligence (AI) in diagnosing BE dysplasia under endocytoscopy (EC). The authors develop diagnostic criteria that can distinguish neoplastic from nonneoplastic Barrett’s based on previous work performed using endocytoscopy on ex-vivo specimens as well as validated CLE criteria. The authors then developed an EC AI model using a ResNet-like CNN architecture. The performance of the criteria and role of the AI model to assist the operators were evaluated in two cohorts of observers using still images. Although the patient sample size is small in this study, it has a decent novelty, and the observer cohort design is robust. The authors make the effort to evaluate the performance of physicians before, after and during follow-up period of the training, which is close to the clinical setting. The authors find that diagnostic accuracy improve when observers were assisted by the AI system. Overall, this is a good study. I am somehow surprised that the authors decided to go in one study, from a clinical trial using a relatively novel technology, to development of diagnostic criteria on video frames to finally develop AI system to assist non-expert physician. This is a large task and perhaps quite ambition. As results, there are some issue to consider.

Major issues:

1.      A major drawback of the study is that the size in test set might be too small to prove the AI’s robustness. As I understand, the author developed the criteria using the videos from the in-vivo study cohort, train the AI model and then validate the model on the images from the same cohort. Ideally the validation of the criteria and of the AI model should be done on a separate cohort of images. This limits greatly the methodology of the study. I understand the data are difficult for collecting in clinic, but would it be possible to use endocytoscopy GI images published in previous open-access papers or books for testing? Even if there are only a few typical cases, it would further improve the credibility of AI testing.

2.      In this study, two things were proved: 1) The potential of identifying BE dysplasia under endocytoscopy ; 2) and AI support makes doctors to perform better in identifying BE dysplasia under endocytoscopy. From the reviewer’s aspect, although the first point was limited by the sample size, the second point was convictive due to its firm design. I would like the authors to expand in discussion what the next steps they envisage to be to take forward the field of endocytoscopy.

3.      How would the score of AI’s prediction affect the decision of endoscopists? To answer this question, can the authors do a sub-analysis by diving the cases into different groups with different levels of AI scores in the second cohort in order to investigate the impact of the AI score on the physician diagnosis? In other words, would a high AI scope more likely to influence the physicians diagnosis?

4.      In cohort 2, why the authors provided optional call to AI support instead of automatic predictions from AI. Does it mean endoscopist should press a button to get the answer during endoscopy when the AI applied into clinical setting? Please explain.

5.      As I understand, a pathologist was not involved in the definition of the EC criteria. This is a limitation to be acknowledged, as a pathologist would provide a significant insight on the description and interpretation of the patterns and credibility of the diagnostic criteria.

6.      Can the authors be more clear on who and how annotated the individual frames in the test set? Given the dynamic nature of a video, multiple frames from the same video taken on a dysplastic lesion for example could show different stages of disease including non-neoplastic disease. Were all images from a neoplastic lesion considered bona-fide neoplastic?

Minor issues:

1.      The baseline information of enrolled endoscopists and gastroenterologists should be clarified, for example, their endoscopic experience, if they are novices/seniors/experts, etc. The levels of doctors may affect the clinical significance of the result.

2.      Was the experience of gastroenterologists in the two groups in cohort 2 matched?

3.      Introduction: Ref 3 does not apply to this statement as this is a RCT to investigate different techniques of endoscopic resection

4.      Introduction: Sensitivity of Seattle protocol has been recently assessed in a cross-over RCT doi: 10.1016/j.cgh.2022.01.060

5.      Introduction: there is virtually no mention of endocytoscopy in Barrett’s. It is true that the work done is limited and but it would be important to refer in the main text (and not only inro supplementary material) the papers from Pohl et al and Tomizawa et al

6.      Methods line 130: what does randomly mean in this context?

7.      Methods line 131: could the authors explain the difference between endoscopists and gastroenterologists, as the two group can be interchangeable?

8.      Methods: In the second cohort of gastroenterologists, why was the AI group not crossed over to non-AI after a wash out period?

9.      Can the author provide a more explanation in discussion about how endocytoscopy  compares with CLE? Does CE offers advantages compared to CLE, which despite years of enthusiasm has not been implemented into to routine clinical work?

Author Response

Reviewer 1

This is an interesting study exploring the possibility of applying intelligence (AI) in diagnosing BE dysplasia under endocytoscopy (EC). The authors develop diagnostic criteria that can distinguish neoplastic from nonneoplastic Barrett’s based on previous work performed using endocytoscopy on ex-vivo specimens as well as validated CLE criteria. The authors then developed an EC AI model using a ResNet-like CNN architecture. The performance of the criteria and role of the AI model to assist the operators were evaluated in two cohorts of observers using still images. Although the patient sample size is small in this study, it has a decent novelty, and the observer cohort design is robust. The authors make the effort to evaluate the performance of physicians before, after and during follow-up period of the training, which is close to the clinical setting. The authors find that diagnostic accuracy improve when observers were assisted by the AI system. Overall, this is a good study. I am somehow surprised that the authors decided to go in one study, from a clinical trial using a relatively novel technology, to development of diagnostic criteria on video frames to finally develop AI system to assist non-expert physician. This is a large task and perhaps quite ambition. As results, there are some issue to consider. 

We would like to thank the reviewer for the time that was invested in reading our manuscript as well as the expressed compliments and the valuable input for improving our manuscript. Below, we address the issues that have been expressed by the reviewer.

Major issues:

  1. A major drawback of the study is that the size in test set might be too small to prove the AI’s robustness. As I understand, the author developed the criteria using the videos from the in-vivo study cohort, train the AI model and then validate the model on the images from the same cohort. Ideally the validation of the criteria and of the AI model should be done on a separate cohort of images. This limits greatly the methodology of the study. I understand the data are difficult for collecting in clinic, but would it be possible to use endocytoscopy GI images published in previous open-access papers or books for testing? Even if there are only a few typical cases, it would further improve the credibility of AI testing. 

Thank you for pointing out this issue and providing us the opportunity to clarify it. The diagnostic criteria for differentiating neoplastic from nonneoplastic Barrett’s tissue were indeed based on previous work performed using endocytoscopy in BE as well as validated CLE criteria.

However, for training and (external) testing the AI model the cohort was split in two independent separate cohorts to prevent overfitting in the parameters of the AI model. Thus, one part of the cohort concerned data that was solely used for training and internal validation; and the other part of the cohort was used for (external) testing the AI algorithm. In other words, data was solely used for either training and internal validation or for testing. Thus, the results of the AI algorithm in the (external) testing dataset are presented in the manuscript. We have clarified this in paragraph 2.3 Training, validating, and testing of the CNN architecture (from line 186 onwards).

We appreciate the recommendation of the reviewer to search for open-access datasets to test the AI performance; however the articles that involve EC in BE are limited to the work by Tomizawa et al (2013) and Pohl et al. (2007). These studies employed a previous EC model, namely the probe-based XEC120, Olympus Medical Systems Co., Tokyo, Japan) with a magnification factor x1100 in Tomizawa et al., and the probe-based model with a x1125 and x450 magnification factor in the article by Pohl et al. In our study, we used the integrated EC version (GIF-H290 EC) with a 520-fold magnification factor. Furthermore, the used another staining protocol (solely methylene blue) than we did (mixture of methylene blue and crystal violet). Other papers investigating AI in EC, concern squamous-type dysplasia and carcinoma of the esophagus, colorectal polyps/cancer, or inflammatory bowel disease. Thus, conditions that insufficiently correspond to BE on a histologic, and so EC basis, that could be used for reliable external testing of the algorithm trained in our cohort. If more EC studies are conducted the future, the proposed approach by the reviewer would be indeed very interesting to conduct.

  1. In this study, two things were proved: 1) The potential of identifying BE dysplasia under endocytoscopy ; 2) and AI support makes doctors to perform better in identifying BE dysplasia under endocytoscopy. From the reviewer’s aspect, although the first point was limited by the sample size, the second point was convictive due to its firm design. I would like the authors to expand in discussion what the next steps they envisage to be to take forward the field of endocytoscopy.

We thank the reviewer for this positive remark. We have elaborated in our discussion the next steps we envisage to move the field of endocytoscopy forward (lines 357-368).

  1. How would the score of AI’s prediction affect the decision of endoscopists? To answer this question, can the authors do a sub-analysis by diving the cases into different groups with different levels of AI scores in the second cohort in order to investigate the impact of the AI score on the physician diagnosis? In other words, would a high AI scope more likely to influence the physicians diagnosis?

We thank the reviewer for this interesting suggestion. In addition to the lines that we wrote in paragraph 3.5 Man-machine interaction, we have elaborated on the interesting tendencies that were observed (lines 322-326). In addition, see also our response below number 4.  

  1. In cohort 2, why the authors provided optional call to AI support instead of automatic predictions from AI. Does it mean endoscopist should press a button to get the answer during endoscopy when the AI applied into clinical setting? Please explain.

In this study, we provided AI as a second assessor to the gastroenterologists. This application means that gastroenterologists indeed had the option to call for AI-assistance in case they were in doubt of a specific image, providing the AI prediction up front could potentially bias the gastroenterologist. Currently, it is thought that AI in GI endoscopy will not replace the operator but rather assist and augment the operator’s skills. Through the utilization of AI as second assessor in our study, we were able to assess the man-machine collaboration in which an endoscopist can establish unbiased a diagnosis and consult the algorithm when considered necessary. We think future studies should investigate which workflow works best during clinical procedures.

  1. As I understand, a pathologist was not involved in the definition of the EC criteria. This is a limitation to be acknowledged, as a pathologist would provide a significant insight on the description and interpretation of the patterns and credibility of the diagnostic criteria. 

During the study, a GI expert pathologist (AK) was involved in achieving consensus regarding the EC criteria for EC metaplasia and EC neoplasia for which we conducted a final meeting (Supplementary Material: EC image classification, paragraph 3. Design of the endocytoscopic BE classification system, line 90). We involved him in his study due to his expertise on BE as well as his knowledge of ultra-magnification endoscopy, i.e. CLE. We now address this more specifically in line 90. 

  1. Can the authors be more clear on who and how annotated the individual frames in the test set? Given the dynamic nature of a video, multiple frames from the same video taken on a dysplastic lesion for example could show different stages of disease including non-neoplastic disease. Were all images from a neoplastic lesion considered bona-fide neoplastic?

We acknowledge this valid issue from the reviewer and we addressed this in Supplementary Material 1: In vivo EC imaging procedure (lines 33-35). If a lesion was present, we first videotaped an adjacent, macroscopically non-aberrant spot and afterwards the lesion to potentially facilitate an endocytoscopic comparison within a Barrett segment between non-suspected and suspected tissue. During endocytoscopy, we attached a transparent cap at the distal tip of the endoscope which enabled the operator (WN, JW, FP) to apply some suctioning power that would maintain EC-tissue contact at one spot.

Minor issues:

  1. The baseline information of enrolled endoscopists and gastroenterologists should be clarified, for example, their endoscopic experience, if they are novices/seniors/experts, etc. The levels of doctors may affect the clinical significance of the result.

We thank the reviewer for pointing out this missing information and have provided baseline information of the enrolled endoscopists and gastroenterologists below. However, we expect this information minimally affects the results as EC generates a completely new view on BE tissue in comparison to conventional endoscopy due to 1) the 520-magnification factor visualization 2) the tissue staining with methylene blue and crystal violet.

Cohort 1

Cohort 2

Gastroenterology residents

5

0

Gastroenterologists with expertise in BE from the Dutch BE expert centers

5

0

Gastroenterologists from academic center

0

9

Gastroenterologists from community hospitals

0

3

Experience in Endocytoscopy

0

0

Experience in AI

2

0

  1. Was the experience of gastroenterologists in the two groups in cohort 2 matched?

Both groups of gastroenterologists from cohort 2 involved Dutch gastroenterologists with experience in BE surveillance but without experience in EC as well as the endoscopic application of AI.

  1. Introduction: Ref 3 does not apply to this statement as this is a RCT to investigate different techniques of endoscopic resection.

We have removed this reference.

  1. Introduction: Sensitivity of Seattle protocol has been recently assessed in a cross-over RCT doi: 10.1016/j.cgh.2022.01.060

We have added this reference (line 60).

  1. Introduction: there is virtually no mention of endocytoscopy in Barrett’s. It is true that the work done is limited and but it would be important to refer in the main text (and not only inro supplementary material) the papers from Pohl et al and Tomizawa et al 

As pointed out by the reviewer, the work done of EC in BE is limited and solely reported in the work by Pohl et al and Tomizawa et al. Following the reviewer’s suggestion, we have added lines regarding this topic in the introduction (lines 69-71).

  1. Methods line 130: what does randomly mean in this context?

We invited a random cohort of gastroenterologists that are employed in the north of the Netherlands to participate in the online modules (which was a cohort > 12 gastroenterologists). The first gastroenterologists that responded to the invitation of participation were included.

  1. Methods line 131: could the authors explain the difference between endoscopists and gastroenterologists, as the two group can be interchangeable?

We chose this nomenclature to ensure distinction of the two cohorts throughout the manuscript.

  1. Methods: In the second cohort of gastroenterologists, why was the AI group not crossed over to non-AI after a wash out period?

      This would have been interesting indeed. However, our study primarily focused on the potential of AI-assistance in endocytoscopy.

  1. Can the author provide a more explanation in discussion about how endocytoscopy compares with CLE? Does EC offers advantages compared to CLE, which despite years of enthusiasm has not been implemented into to routine clinical work?

We have added our considerations regarding this issue in the discussion section in lines 357-365.

Reviewer 2 Report

The authors investigated whether artificial intelligence (AI) as second assessor can assist doctor in assessing complex, microscopic endoscopy images on the presence of esophageal cancer. They developed online training and testing modules for doctors to learn to classify these novel images, and to assess the potential of AI-assistance in analyzing the esophageal microscopy images. They showed that the best diagnostic scores for cancer recognition emerged through the collaboration between doctors and AI as second assessor. They concluded that AI could be used to support clinical implementation of endoscopy techniques that generate microscopic images. This manuscript is worth being published in this Cancers and gives us instructive message. However, as they pointed out in the discussion part as a limitation, the data set was relatively small, so that future clinical implementation of optical biopsy by advanced endoscopy techniques would be tested.

Author Response

Reviewer 2

The authors investigated whether artificial intelligence (AI) as second assessor can assist doctor in assessing complex, microscopic endoscopy images on the presence of esophageal cancer. They developed online training and testing modules for doctors to learn to classify these novel images, and to assess the potential of AI-assistance in analyzing the esophageal microscopy images. They showed that the best diagnostic scores for cancer recognition emerged through the collaboration between doctors and AI as second assessor. They concluded that AI could be used to support clinical implementation of endoscopy techniques that generate microscopic images. This manuscript is worth being published in this Cancers and gives us instructive message. However, as they pointed out in the discussion part as a limitation, the data set was relatively small, so that future clinical implementation of optical biopsy by advanced endoscopy techniques would be tested. 

We would like to thank the reviewer for taking the time for reading our manuscript, the expressed appreciation for our work, and subsequently for the positive recommendation.

Reviewer 3 Report

Authors describe the use of artificial intelligence to assist endoscopists and gastroenterologist to evaluate endoscopic images. Authors describe their hypothesis in the simple summary where they state that endoscopic images are highly detailed and difficult for doctors to interpret, but when images are “too complex for human interpretation” artificial intelligence could be a useful tool to assist doctors. Whilst the suggested causal relation between “difficult for doctors to interpret” and “too complex for human interpretation” can come over as a bit presumptuous, authors refine their hypothesis in the following abstract. The manuscript is well written but could benefit from some clarification regarding the following:

Authors describe the substantial difference between “expert” and non-expert evaluators of dysplasia and the application of AI in Barrett dysplasia in ref 22. In this manuscript authors describe this module and its application to assist non-experts. Do endoscopist and gastro-enterologists both evaluate these images in regular diagnostic procedures and if so, what would be the output of “expert” endoscopist assisted optical biopsy evaluation (or vice versa, “expert” gastro-enterologist assisted optical biopsy evaluation). Would regular training by a “human “ be just as helpful and the results more related to training evaluators and not in the assistance of trained gastro-enterologists (line 399)

One group is missing from this overview: unassisted AI evaluation. What are the output measures without any endoscopist or gastroenterologist? This would also be a baseline measurement

Figure 2: Please describe a bit better what is used in these experiments (only the A pat and what is used to train the module (the A and B part?)

There is just a brief mentioning of the AI module used: Residual Encoder-Decoder Capsule -like Convolutional Neural Network architecture. Although the authors referrer to their manuscript in which the describe this module in detail, readers would benefit from more information on how this deep-learning module is used to reconstruct holographic images (e.g. what is the “-like” part?). Does the module recognises and uses specific cell types or structures often missed by non-expert evaluators, can the module recognise structures that are difficult to discriminate by a non-expert evaluator. If yes, would an expert evaluator focuses on these items and have a better yield?

Author Response

Reviewer 3

Authors describe the use of artificial intelligence to assist endoscopists and gastroenterologist to evaluate endoscopic images. Authors describe their hypothesis in the simple summary where they state that endoscopic images are highly detailed and difficult for doctors to interpret, but when images are “too complex for human interpretation” artificial intelligence could be a useful tool to assist doctors. Whilst the suggested causal relation between “difficult for doctors to interpret” and “too complex for human interpretation” can come over as a bit presumptuous, authors refine their hypothesis in the following abstract. The manuscript is well written but could benefit from some clarification regarding the following:

We thank the reviewer for taking time for reading our manuscript, for the positive comments, and for providing us the opportunity to optimize our manuscript.

Authors describe the substantial difference between “expert” and non-expert evaluators of dysplasia and the application of AI in Barrett dysplasia in ref 22. In this manuscript authors describe this module and its application to assist non-experts. Do endoscopist and gastroenterologists both evaluate these images in regular diagnostic procedures and if so, what would be the output of “expert” endoscopist assisted optical biopsy evaluation (or vice versa, “expert” gastro-enterologist assisted optical biopsy evaluation). Would regular training by a “human“ be just as helpful and the results more related to training evaluators and not in the assistance of trained gastro-enterologists (line 399) 

Endocytoscopy is not a regular technique employed for diagnostics in BE. Thus, this concerns a novel application in which currently experts and non-experts cannot be distinguished. We chose this technique in BE, as it holds potential for optical biopsy during endoscopic BE surveillance. As EC displays histological features of the tissue through its 520-fold magnification view, its images incorporate many details. We consider these features to be associated to many current and emerging advanced techniques in endoscopy. Therefore, we hypothesized that AI would have an interesting potential to gastroenterologists in enhancing the recognition of dysplasia on the images of these techniques.

One group is missing from this overview: unassisted AI evaluation. What are the output measures without any endoscopist or gastroenterologist? This would also be a baseline measurement

The output measures of the AI algorithm are ROC curves that were obtained as an average from the five CNN models and resulted in AUCs of 0.894 for test set 1 and 0.923 for test set 2 and can be found in paragraph 2.3 Results of testing the AI algorithm.

In addition, for test set 1, the CNN achieved an 88.0% (81.08-94.92) sensitivity, a 70.67% (56.81-84.52) specificity, and a 79.33% (75.87-82.80) accuracy. For test set 2, the CNN achieved an 82.67% (71.56-93.77) sensitivity, 77.33% (62.29-92.37) specificity, and an 80.0% (77.07-82.93) accuracy.

Figure 2: Please describe a bit better what is used in these experiments (only the A pat and what is used to train the module (the A and B part?)

We thank the reviewer to provide us the opportunity to clarify this issue. The images in panel A reflect the steps of an endoscopic procedure through EC; the EC metaplasia and EC neoplasia images in B for dataset creation. We have added a line to the figure legend.

There is just a brief mentioning of the AI module used: Residual Encoder-Decoder Capsule -likeConvolutional Neural Network architecture. Although the authors referrer to their manuscript in which the describe this module in detail, readers would benefit from more information on how this deep-learning module is used to reconstruct holographic images (e.g. what is the “-like” part?). Does the module recognises and uses specific cell types or structures often missed by non-expert evaluators, can the module recognise structures that are difficult to discriminate by a non-expert evaluator. If yes, would an expert evaluator focuses on these items and have a better yield?

We recognize that the original wording of the CNN architecture is confusing. We have reworded the mentioning to: ‘the decoder of a CNN architecture based on ResNet. The architecture does not reconstruct images to produce a visual output. It merely produces a classification score by processing an input image.

It is generally accepted that CNN architectures make classification decisions based on specific features of an image. This may include specific cell types or structures that non-expert evaluators have trouble detecting. However, CNN’s have a black-box nature, meaning that it is difficult to determine exactly what parts of on image result in a particular classification. There are many researchers currently working on the explainability of these types of models but this is still an ongoing area of research. Currently we are not able to indicate to assessors what to look for in these types of images. The reviewer does rightfully point out that that having access to this information would most likely be very beneficial to assessors.

Round 2

Reviewer 1 Report

With concern to major issue 1, although I appreciate that different image sets are used to train and test the AI system (thanks to the authors to specifying this), I wanted to draw the authors’ attention to the fact that the same image set was used to develop the diagnostic criteria and then develop the AI system. The authors say in the introduction: Therefore, we first prospectively acquired EC frames of BE. We used these frames to develop and validate an endocytoscopic BE classification system in a cohort of ten endoscopists through an online platform, and to train and test a CNN architecture. This generate the possibility that the AI system works well only on this particular cohort (where the diagnostic criteria were developed) and not in external cohorts. This needs to be acknowledged in the limitations of the study

Minor Issue number 1: I would recommend to add the table in Suppl material

Minor point 6: please modify text explaining how many were originally invited and that recruitment was based on time of response to invitation, so that the term randomly is clear

Minor point 7: the difference between gastroenterologist and endoscopists remain unclear. I appreciate there could be geographical variation, however in general most gastroenterologists are endoscopists and the majority of endoscopists are gastroenterologist. Can the authors address this point more clearly?

Author Response

With concern to major issue 1, although I appreciate that different image sets are used to train and test the AI system (thanks to the authors to specifying this), I wanted to draw the authors’ attention to the fact that the same image set was used to develop the diagnostic criteria and then develop the AI system. The authors say in the introduction: Therefore, we first prospectively acquired EC frames of BE. We used these frames to develop and validate an endocytoscopic BE classification system in a cohort of ten endoscopists through an online platform, and to train and test a CNN architecture. This generate the possibility that the AI system works well only on this particular cohort (where the diagnostic criteria were developed) and not in external cohorts. This needs to be acknowledged in the limitations of the study

We would like to thank the reviewer for taking time to review our manuscript again. We have updated our manuscript and addressed this limitation in line 382 – 387.

Minor Issue number 1: I would recommend to add the table in Suppl material

We have added the table as Supplementary Table 5 in the Supplementary material (lines 136-137).

Minor point 6: please modify text explaining how many were originally invited and that recruitment was based on time of response to invitation, so that the term randomly is clear

 We have extended the description in lines 132 to 145 in the manuscript.

Minor point 7: the difference between gastroenterologist and endoscopists remain unclear. I appreciate there could be geographical variation, however in general most gastroenterologists are endoscopists and the majority of endoscopists are gastroenterologist. Can the authors address this point more clearly?

As the participants of cohort 1 encompassed both gastroenterologists with expertise in BE and residents in gastroenterologists, we refer to this group as endoscopists (see also lines 132 to 145 of the manuscript). As the participants of cohort 2 included solely general gastroenterologists, we refer to this cohort as the gastroenterologists cohort. Moreover, this arbitrary terminology facilitates clarification of the cohorts throughout the manuscript.

Reviewer 3 Report

i have no further suggestions

Author Response

We would like to thank the reviewer for the comments that improved our manuscript and for the positive recommendation.
